# Forest Spectral Recovery and Regeneration Dynamics in Stand-Replacing Wildfires of Central Apennines Derived from Landsat Time Series

**Donato Morresi** [1],*, **Alessandro Vitali** [2], **Carlo Urbinati** [2] **and Matteo Garbarino** [1]

[1] Department of Agricultural, Forest and Food Sciences, University of Torino, Grugliasco (TO), IT 10095, Italy; matteo.garbarino@unito.it

[2] Department of Agricultural, Food and Environmental Sciences, Marche Polytechnic University, Ancona (AN), IT 60121, Italy; alessandro.vitali@univpm.it (A.V.); c.urbinati@univpm.it (C.U.)

* Correspondence: donato.morresi@unito.it; Tel.: +39-011-6705537

**Abstract:** Understanding post-fire regeneration dynamics is an important task for assessing the resilience of forests and to adequately guide post-disturbance management. The main goal of this research was to compare the ability of different Landsat-derived spectral vegetation indices (SVIs) to track post-fire recovery occurring in burned forests of the central Apennines (Italy) at different development stages. Normalized Difference Vegetation Index (NDVI), Normalized Difference Moisture Index (NDMI), Normalized Burn Ratio (NBR), Normalized Burn Ratio 2 (NBR2) and a novel index called Forest Recovery Index 2 (FRI2) were used to compute post-fire recovery metrics throughout 11 years (2008–2018). FRI2 achieved the highest significant correlation (Pearson's $r = 0.72$) with tree canopy cover estimated by field sampling (year 2017). The Theil–Sen slope estimator of linear regression was employed to assess the rate of change and the direction of SVIs recovery metrics over time (2010–2018) and the Mann–Kendall test was used to evaluate the significance of the spectral trends. NDVI displayed the highest amount of recovered pixels (38%) after 11 years since fire occurrence, whereas the mean value of NDMI, NBR, NBR2, and FRI2 was about 27%. NDVI was more suitable for tracking early stages of the secondary succession, suggesting greater sensitivity toward non-arboreal vegetation development. Predicted spectral recovery timespans based on pixels with a statistically significant monotonic trend did not highlight noticeable differences among normalized SVIs, suggesting similar suitability for monitoring early to mid-stages of post-fire forest succession. FRI2 achieved reliable results in mid- to long-term forest recovery as it produced up to 50% longer periods of spectral recovery compared to normalized SVIs. Further research is needed to understand this modeling approach at advanced stages of post-fire forest recovery.

**Keywords:** post-fire recovery; spectral vegetation index; NDVI; NDMI; NBR; NBR2; Integrated Forest *z*-score (IFZ); Forest Recovery Index 2 (FRI2); burn severity; recovery trend modeling

---

## 1. Introduction

The interaction between climate and land-use changes is raising the frequency, surface area, and severity of wildfires in the Mediterranean Basin [1–4]. Among climate change effects, long periods of dry weather are expected to increase fire danger in southern European mountains both under short- and long-term climatic scenarios [5]. The development of appropriate management strategies is essential to prevent fire occurrence and to enhance forest recovery [6,7]. The latter is a critical ecological process after a stand-replacing disturbance, referring to the re-establishment or re-development of forest biomass and canopy structure [8–10]. This process affects regional and global carbon cycles [11,12] and promotes numerous ecosystem services [8]. Furthermore, forest degradation

in the xeric Mediterranean mountains induced by a fire frequency increase can occur even in stands dominated by fire-adapted tree taxa [7]. Forest recovery at a landscape scale is often modeled as a fast and homogeneous process. However, depending on the fire severity level, it might be a very diversified one [9], with great changes in forest structure and species composition [13].

Since past decades, satellite optical remote sensing was widely adopted for the analysis of post-fire forest recovery [14]. Imagery collected by the United States (US) Landsat program is currently considered as the most valuable source of time-series data at a landscape scale [8,15]. This is primarily due to the long-term availability of systematically acquired images, spanning over 40 years. The unique combination of 30-m spatial resolution and 16 days of revisiting time enables assembling conspicuous Landsat time series (LTS) [16]. Monitoring forest regeneration development through gradual changes in the optical spectral domain is challenging, given the confounding effects from a variety of factors such as phenology and sun angles [15,17]. To reduce these types of noise, selecting near-anniversary acquisition dates [17] or using unburned neighboring pixels as control areas [18,19] were proposed. However, limitations such as data gaps in the time series [15] can negatively affect outcomes. Recent development in dense Landsat time-series collection by including all the clear yearly observations [20] proved to be useful in characterizing both intra-annual variations and long-term temporal trajectories [15].

Spectral vegetation indices (SVIs) [21–24] and spectral mixture analysis (SMA) [25–27] are common remote-sensing techniques to track post-fire vegetation recovery using LTS, and were adopted in different vegetation communities and ecosystems [14,28]. SVIs from LTS are widely used since they can maximize the sensitivity to plant biophysical factors and reduce the noise from atmosphere, landforms, and soil variability [29,30]. Specifically, Normalized Difference Vegetation Index (NDVI) was widely employed to assess post-fire vegetation recovery using a multi-temporal approach in several burned sites of the Mediterranean Basin, primarily because of the high correlation achieved with field measurements such as fractional vegetation cover [27,31]. Other studies in North American boreal forests focused on indices contrasting the near-infrared (NIR) and shortwave-infrared (SWIR) bands of Landsat TM/ETM+/OLI sensors to track post-fire recovery [23,24,32]. Since the shortwave-infrared region of the spectral domain is sensitive to variations in the forest structure [17,33,34], Normalized Difference Moisture Index (NDMI) [35] and Normalized Burn Ratio (NBR) [36] or the SWIR1 band alone [24,37] are of great interest to monitor post-disturbance forest recovery. Additionally, Normalized Burn Ratio 2 (NBR2) takes advantage of the contrast between the two Landsat sensor SWIR bands, with promising results in the assessment of post-fire vegetation recovery in the shrublands of California [38] and sclerophyll forests of Australia [39].

The Integrated Forest *z*-score (IFZ) [40,41] is a threshold-based index that was initially developed as a part of the Vegetation Change Tracker algorithm [40] in order to target abrupt forest cover changes at the pixel level. Specifically, IFZ is an inverse measure of the likelihood of a pixel to be forested, which is obtained by computing its spectral distance from defined forest pixels. Some authors proposed using the reciprocal of IFZ, termed Forest Recovery Index (FRI), to allow for the comparison with other spectral indices growing in direct proportion with the amount of vegetation cover such as NDVI [41]. To date, few studies employed either IFZ or FRI in post-fire forest recovery tracking [41–43], but its potential toward the detection of long-term forest recovery dynamics was highlighted within ponderosa pine forests [42], boreal larch forests [41], and a mosaic of mixed conifer forests in the Greater Yellowstone Ecosystem [43].

Post-fire recovery rates were assessed through trend analysis on LTS which involved fitting linear, non-linear [25], and segmented [44] pixel-wise models to near-anniversary date images to characterize the spatial variability of this gradual process. This approach proved to be effective, but trend analysis based on single curve fitting can be biased by outliers [16], making the adoption of robust regression models preferable.

Another challenging factor to be considered is the land-cover heterogeneity due to the anthropogenic disturbances. This is particularly evident in human-shaped landscapes featuring a complex patch mosaic of crops, forests, pastures, and human infrastructures.

The present study aimed to compare the efficiency of different spectral vegetation indices to assess the early forest recolonization patterns in four burned landscape mosaics of the central Apennines (Italy). Wildfires in this mountain ecoregion are one of the most common natural disturbances [45,46], but forest recovery dynamics is scarcely studied with a remote-sensing approach. Our general hypotheses to be tested were as follows: (a) Landsat-derived spectral vegetation indices employing the SWIR bands have enhanced sensitivity toward post-fire forest recovery dynamics; (b) forest regeneration processes under different burn severity degrees and forest types can be inferred from recovery patterns of spectral vegetation indices; (c) the ability of Landsat-derived spectral vegetation indices to track diachronic post-fire forest recovery dynamics can be assessed through spatially explicit robust regression models.

## 2. Materials and Methods

### 2.1. Study Areas

The study was held in the central Apennines and it included areas of the Marche and Abruzzo administrative regions. Four large stand-replacing wildfires were located using MODIS Collection 6 Level 3 monthly burned area products (MCD64A1) [47,48]. The correspondent study areas were named with the nearest municipality: Roccafluvione (RF), L'Aquila (LA), Navelli (NA), and Roccamorice (RM) (Figure 1). This dataset also provided the starting date of each wildfire and the overall burned surface area (Table 1). The extent of forest areas affected by wildfires was initially estimated by intersecting the Corine Land Cover 2006 (CLC) forest cover map (codes 311, 312, 313) [49] with the MODIS burned area products in a GIS environment (Table 1).

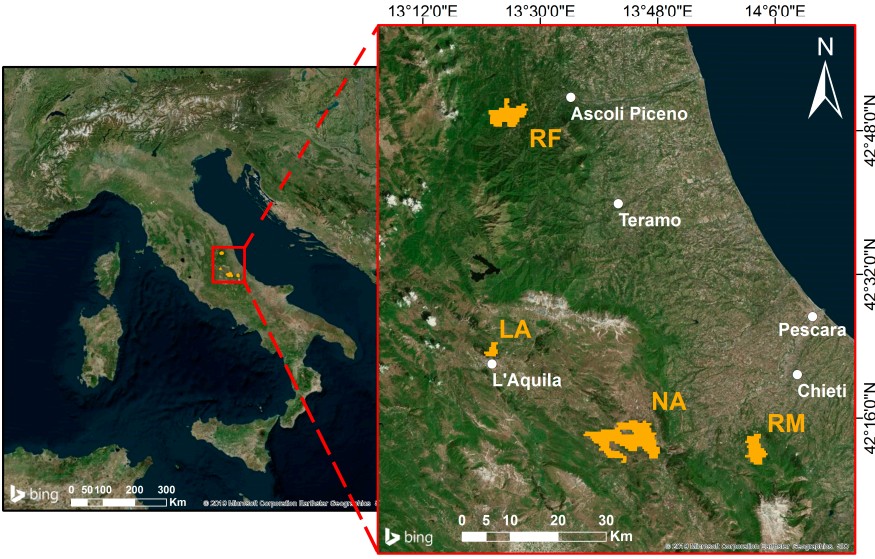

**Figure 1.** Location and wildfire surface (orange polygons) from MODIS burned area (MCD64A1) of the four study areas: Roccafluvione (RF), L'Aquila (LA), Navelli (NA), and Roccamorice (RM).

The large wildfires which occurred in central and southern Italy during the summer of 2007 were driven by severe climate conditions, similar to those arising in Greece during the same year [50,51]. The occurrence of previous prolonged drought periods, the high summer temperatures, and strong winds enhanced the spread of large wildfires [50]. Other environmental and social factors contributed to raising the fire risk in the study areas, such as the abandonment of agricultural lands and the lack of regular fire prevention forest management [50]. Moreover, the suppression of these wildfires was made difficult due to their almost synchronic occurrence (same days of July and August) [50]. The impacted stands were mainly conifer plantations pure or mixed with indigenous broadleaved woodlands classified into three different forest types of the regional forest inventories. Hardwood

stands were dominated by pubescent oak (*Quercus pubescens*) (Po) or by mixed manna ash (*Fraxinus ornus*) and European hop-hornbeam (*Ostrya carpinifolia*) (Ma). European black pine (*Pinus nigra*) (Pp) dominated conifer plantations. Climate and landform similarities between the study areas (Table 1) were highlighted to support the comparison of post-fire recovery dynamics between these sites. Climatic data were extracted from WorldClim Version 2 grids with a spatial resolution of ~1 km² [52], whereas geomorphological features were derived from the 10-m resolution TINITALY DEM [53].

**Table 1.** Wildfire information, and climate and landform properties of the four study areas: Roccafluvione (RF), L'Aquila (LA), Navelli (NA), and Roccamorice (RM).

|  | RF | LA | NA | RM |
|---|---|---|---|---|
| Fire start date | 21 July 2007 | 9 August 2007 | 14 July 2007 | 23 July 2007 |
| Overall burned area (hectares) | 2753 | 530 | 6939 | 1823 |
| Forest burned area (hectares) | 1860 | 391 | 1896 | 427 |
| Annual mean temperature (°C) | 12.4 | 11 | 11.8 | 10.9 |
| Annual mean precipitation (mm) | 820.9 | 856.7 | 827.1 | 743.9 |
| Mean altitude (m) $\pm$ SD | 628 $\pm$ 146 | 977 $\pm$ 104 | 787 $\pm$ 206 | 933 $\pm$ 206 |
| Mean slope (°) $\pm$ SD | 28 $\pm$ 9 | 21 $\pm$ 8 | 17 $\pm$ 9 | 17 $\pm$ 9 |
| Mean roughness index | 23 | 12 | 9 | 10 |
| Heat Load Index $\pm$ SD | 0.76 $\pm$ 0.09 | 0.79 $\pm$ 0.05 | 0.74 $\pm$ 0.06 | 0.76 $\pm$ 0.07 |

## 2.2. Dataset and Preprocessing

Forest regeneration dynamics were explored using a series of Landsat annual image composites at one-year intervals between 2001 and 2018, resulting in a total period of 18 years (seven pre-fire and 11 post-fire) for each study area. Annual composites were produced with priority to images acquired closest to the day of peak (DOP) of the same growing season and with the highest percentage of valid pixels (i.e., not contaminated by clouds/cloud shadows or with missing values). The selected DOP corresponds to 12 July in non-leap years (day-of-year 197) and was derived from the EVI2 Long-Term Average Phenology (from 1980 to 2010) available at the Vegetation Index and Phenology Lab Data Explorer [54], which contains historical annual phenology parameters obtained from AVHRR and MODIS sensors for homogeneous vegetation clusters [55]. The least cloud-contaminated acquisitions available for the study areas in the growing season (1 June–31 August) were selected to produce annual composites. The dataset includes Landsat TM, ETM+, and OLI images acquired in the WRS-2 Path/Row 190/30, 190/31, and 191/30 (details are provided in Table S1, Supplementary Materials). The majority of Landsat data was provided by the USGS Earth Resources Observation and Science (EROS) Center Science Processing Architecture (ESPA) On-Demand Interface [56] processed in surface reflectance (Level-2 Science Products). Surface reflectance products of Landsat 5 TM and Landsat 7 ETM+ were generated using the Landsat Ecosystem Disturbance Adaptive Processing System (LEDAPS) [57]. Instead, Landsat 8 OLI processing was based on the Landsat Surface Reflectance Code (LaSRC) [58]. Two Landsat 5 TM scenes acquired in 2008 (WRS-2 Path/Row 190/31) were available only in the ESA Landsat archive [59] at Level-1T (radiometrically calibrated and orthorectified using ground control points and DEM). They were co-registered with USGS scenes using the tool AROSICS [60] and further converted to surface reflectance using the 6S radiative transfer model [61] employed by LEDAPS and implemented in GRASS GIS 7.2 [62,63]. Clouds and cloud shadows were masked with the function of mask method [64,65]. As Landsat 8 OLI images were about one-third of all images used in this study, they were calibrated to those acquired by Landsat 7 ETM+ through gain and offset coefficients [66].

## 2.3. Fire Perimeter and Burn Severity Assessment

Fire perimeter and the spatial distribution of burn severity patterns within each study area were assessed using the Relative difference Normalized Burn Ratio (RdNBR) [67]. Single Landsat TM summer images of 2006 and 2008 were employed to compute RdNBR as described by Equations (1)–(3). Table S2 (Supplementary Materials) provides detailed information on these images.

$$NBR = \frac{(NIR - SWIR2)}{(NIR + SWIR2)} \tag{1}$$

$$dNBR = \left(\left(NBR_{prefire} - NBR_{postfire}\right) * 1000\right) - dNBR_{offset} \tag{2}$$

$$RdNBR = \frac{dNBR}{\sqrt{\left|NBR_{prefire}\right|}} \tag{3}$$

The evaluation of spectral changes caused by the fire on forest ecosystems during the following vegetative season is defined as an extended assessment of remotely sensed burn severity and included first- and second-order effects caused by fire [68–70]. Because burn severity assessment was not performed through field surveys, the thresholds defined by Miller and Thode [67] were used to define burn severity classes (low, moderate, and high) of the RdNBR. The dNBR offset was computed by averaging dNBR values within undisturbed forest pixels which were delineated for each study area (Section 2.5) to minimize changes in reflectance not caused by fire [67,68,71]. Fire perimeters of each study area were corrected through on-screen digitization of burned pixels in post-fire Landsat TM images (2008) using false-color composites (RGB = SWIR2, NIR, Red), as misclassification between unchanged and low-severity pixels occurred frequently.

### 2.4. Area of Interest within Fire Perimeters

Regeneration dynamics were investigated in those groups of burned pixels containing at least 10% of tree canopy fractional cover before fires and that were larger than 0.5 ha. These parameters were drawn from the definition of forest provided by the Food and Agriculture Organization (FAO) of the United Nations [72]. Pre-fire tree canopy fractional cover within each fire perimeter was computed using pre-fire forest/non-forest land-cover maps obtained from the classification of high-resolution (0.5 m) RGB aerial orthophotos provided by the Italian Agency for payments in agriculture (AGEA) (Table 2) through an object-oriented classification approach using Trimble eCognition Developer software. Firstly, a multi-resolution bottom-up segmentation [73,74] was applied in order to aggregate groups of tree canopies by repeatedly increasing the scale factor toward the stand scale. Secondly, the support vector machine (SVM) classifier [75,76] was applied to the coarser-scale objects using the radial basis function (RBF) kernel with tuning parameters "cost" (C) and gamma proposed by Qian et al. [77]. On-screen validation of 200 randomly distributed points per forest/non-forest class (400 points for each study area) was performed using orthophotos as ground truth reference (Table 2).

**Table 2.** Italian Agency for payments in agriculture (AGEA) orthophoto acquisition dates, accuracy assessment metrics of forest/non-forest cover maps, and percentage of forest cover within fire perimeters. Accuracy metrics: producer's accuracy (PA), user's accuracy (UA), overall accuracy, and K statistic values for each study area (Roccafluvione—RF, L'Aquila—LA, Navelli—NA, Roccamorice—RM).

| | **RF** | **LA** | **NA** | **RM** |
|---|---|---|---|---|
| Acquisition dates | 18 June 2007 | 14 May 2007 9 September 2007 | 14 May 2007 18 June 2007 | 21 June 2007 9 July 2007 |
| PA forest cover (%) | 95.68 | 99.45 | 97.08 | 94.44 |
| PA non-forest cover (%) | 89.3 | 91.71 | 93.46 | 93.56 |
| UA forest cover (%) | 88.5 | 91 | 93.2 | 93.5 |
| UA non-forest cover (%) | 96 | 99.5 | 97.2 | 94.5 |
| Overall accuracy (%) | 92.25 | 95.25% | 95.2% | 94% |
| Kappa coefficient | 0.845 | 0.905 | 0.904 | 0.88 |
| Classified forest cover (%) | 85.75 | 73.38 | 39.11 | 32.68 |

*2.5. Spectral Vegetation Indices*

Post-fire temporal trajectories of burned forests were assessed using five spectral vegetation indices (SVI): NDVI (Normalized Difference Vegetation Index) [78], NDMI (Normalized Difference Moisture Index) [35], NBR (Normalized Burn Ratio) [36], NBR2 (Normalized Burn Ratio 2) [38,39,79], and FRI2 (Forest Recovery Index 2), which is a revised version of FRI (Forest Recovery Index) [41]. They were used with Landsat imagery both for the detection of forest disturbances including fires and to monitor post-disturbance forest dynamics. NDVI and NDMI, and NBR and NBR2 were computed as shown in Equations (4) and (5), and Equation (1) and (6), respectively.

$$NDVI = \frac{(NIR - Red)}{(NIR + Red)} \tag{4}$$

$$NDMI = \frac{(NIR - SWIR1)}{(NIR + WIR1)} \tag{5}$$

$$NBR = \frac{(SWIR1 - SWIR2)}{(SWIR1 + SWIR2)} \tag{6}$$

Like FRI, FRI2 (Equation (7)) is the reciprocal of IFZ [80] (Equation (8)). Adding 1 to IFZ at the denominator avoided obtaining wild values when IFZ was close to 0 and constrained FRI2 to the range between 0 and 1.

$$FRI2 = \frac{1}{(IFZ + 1)} \tag{7}$$

$$IFZ = \sqrt{\frac{1}{NB} \sum_{i=1}^{N} \left( \frac{b_i - \bar{b}_i}{SD_i} \right)^2} \tag{8}$$

In Equation (8), $b_i$ is the spectral value of the pixel in band $i$, $\bar{b}_i$ and $SD_i$ are the mean and standard deviation obtained from forest samples in band $i$, and NB is the number of spectral bands. The Red, SWIR1, and SWIR2 Landsat bands were employed due to their sensitivity to forest cover changes [40]. Yearly means and standard deviations of forest cover were extracted from a forest mask with boundaries outlined by increasing fire perimeter extents of 4 km (Euclidean distance). Specifically, this mask was built selecting those Landsat pixels with a tree canopy fractional cover higher than 90% exhibiting a stable behavior over time. Because pre-fire tree canopy fractional cover was available only within the fire perimeters (Section 2.4), it was estimated outside them using a Random Forest model at the Landsat pixel scale. Pre-fire tree canopy cover maps within the fire perimeters were employed to train the model and the Landsat data (six spectral bands acquired during pre-fire dates in 2007 and its derived SVIs) were used as predictor variables. At last, stable forest pixels were outlined by selecting those with an NBR maximum range lower than 0.15 for the entire analysis interval (2001–2018).

*2.6. Field Data and SVI Correlation*

Tree canopy fractional cover including both dominant and overtopped trees was visually assessed in the field (study area RF) during the summer of 2017 (June and July) using 38 circular plots with 30-m diameter. Centroids of the plots were located close to the center of each Landsat image pixel using a Trimble Juno 3B handheld GPS and a Trimble Pro 6T GNSS receiver having sub-metric horizontal accuracy. SVIs values were extracted using a bilinear interpolation method to limit mismatches between Landsat pixels centroids location and field plots as suggested by Parks et al. [71]. Tree canopy fractional cover was correlated with the SVIs obtained from the image composite of 2017 using Pearson's correlation test.

### 2.7. Post-Fire Recovery Metrics and Temporal Trajectories

A Relative Difference SVI (RDSVI) index was computed for each post-fire SVI as shown in Equation (9), using an algorithm similar to the ones proposed for the burn severity detection [67,71]. Forest spectral recovery causes a decrease in RDSVI values through time since the difference between pre-fire and post-fire decreases as well. The relativization of SVIs allowed for the comparison between recovery dynamics occurring under different ecological conditions such as pre-fire canopy cover density and different forest types. The median pixel value from 2001 to 2007 for each SVI was taken as reference for the pre-fire condition. The averaged difference between pre-fire median and annual post-fire SVIs was extracted from undisturbed forest cover (Section 2.4) to account for inter-annual changes of SVIs. These changes can be attributed to external factors such as phenology and sun angle, similarly to the offset employed for burn severity assessment (Section 2.3). This offset was applied to normalized SVIs (NDVI, NDMI, NBR, and NBR2) as FRI2 is already obtained using yearly spectral values of undisturbed forest cover.

$$RDSVI = \frac{\left(\left(SVI_{prefire\ median} - SVI_{nth\ postfire\ year}\right) - SVI_{offset}\right)}{SVI_{prefire\ median}} \tag{9}$$

Post-fire forest spectral trajectories were assessed by averaging RDSVI values extracted from a set of sampling points located in different burn severity classes and forest types within the area of interest. These points were randomly distributed at a minimum distance of 200 meters in order to reduce the influence of spatial autocorrelation. Global Moran's I test was performed using incremental distances to determine this lowest one. The number of sampling points varied among study areas from 491 to 77, according to the extent of each wildfire.

### 2.8. Statistical Analysis of Recovery Trends

The non-parametric Theil–Sen (TS) slope estimator of linear regression [81,82] was employed to assess pixel-wise changes of RDSVI occurring within the area of interest from 2010 to 2018. The time frame for the analyses started on the third year since fire occurrence. The early post-fire succession in Mediterranean ecosystems usually features a prompt colonization of annual herbs and perennial woody shrubs [83]. This could yield large increases of SVIs [18,21,25], biasing the trend of forest spectral recovery. The TS slope estimator was chosen as it is insensitive to up to 29% of outliers [84,85] and it proved to be effective in detecting SVI trends of forest ecosystems [85–87]. This method involves computing the median of all the slopes between observation values at all pairwise time steps for a total of n(n − 1)/2 slopes. Equation (10) displays how it is computed for observations $Y_j$ and $Y_i$ taken at time $t_j$ and $t_i$.

$$TS\ slope = median\left(\frac{Y_j - Y_i}{t_j - t_i}\right);\ i < j,\ t_i \neq t_j \tag{10}$$

The intercept of the linear trend was computed with the Conover Equation (11) [88],

$$intercept = median(Y) - TS\ slope \times median(t) \tag{11}$$

where median($Y$) and median($t$) are the medians of observations ($Y$) and of the time-series length ($t$). The significance of the TS slope is commonly tested using the rank-based Mann–Kendall (MK) test [84,89–91] through which the existence of a monotonic trend is evaluated, without any assumption regarding its shape. The direction and the power of a monotonic trend is expressed by Kendall's rank correlation coefficient (tau) (Equation (12)) [89,91],

$$\tau = \frac{2S}{n(n-1)} \tag{12}$$

where $-1 \leq \tau \leq 1$. The test statistic $S$ proposed by Mann [89] depends on a series of $n$ repeated observations taken over equal time intervals (Equation (13)).

$$S = \sum_{i=1}^{n-1} \sum_{j=i+1}^{n} \begin{cases} -1, & if\ Y_j - Y_i < 0 \\ 0, & if\ Y_j - Y_i = 0 \\ 1, & if\ Y_j - Y_i > 0 \end{cases} \tag{13}$$

The MK test assumes the observations to be a set of statistically independent variables [84,92] and the presence of serial correlation in SVI time series can lead to overestimating the portion of significant trends [84,93]. Because it appeared to be likely that changes in SVIs observed in Landsat time series are influenced by the underlying post-fire forest recovery process, it is worth considering the existence of a lag-one positive serial correlation (e.g., high observations may tend to follow high observations). Lag-one serial correlation was, thus, removed prior to applying the MK test using the trend-free pre-whitening procedure described in Yue et al. [91] and implemented in the R (R Core Team 2018) package "zyp" [94]. Regression coefficients were used to predict the time required by each SVI to return to its pre-fire spectral values by setting the value of RDSVI equal to zero. Only those pixels displaying jointly a significant negative monotonic trend ($\alpha$-level < 0.01) in all of the SVIs in the MK test were considered reliable to assess recovery times.

## 3. Results

### 3.1. Relationship between Field Data and Landsat-Derived SVIs

The Pearson's correlation test, employed to explore the linear relationship between tree canopy fractional cover available from field surveys and Landsat SVIs, produced slightly different values depending on the SVI (Figure 2). NDVI and FRI2 attained the lowest (0.66) and the highest (0.72) values of Pearson's $r$, respectively, whereas NDMI, NBR, and NBR2 showed identical results.

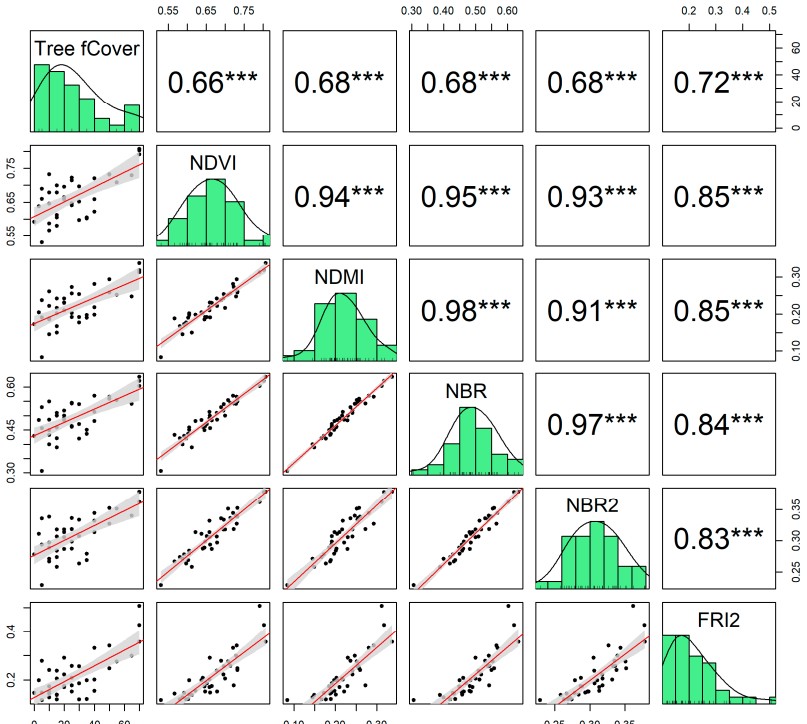

**Figure 2.** Correlation matrix of Pearson's test between tree fractional canopy cover (Tree fCover) obtained from field data and Landsat spectral vegetation indices (SVIs). Asterisks indicate that all tests were significant ($p < 0.001$).

### 3.2. Temporal Trajectories of Post-Fire RDSVIs

Post-fire temporal trajectories of RDSVIs displayed that forest spectral recovery occurred as the difference of each SVI with its pre-fire value was reduced through the years (Figure 3). Different recovery patterns occurred between SVIs (rows in Figure 3) and between study areas, burn severity classes, and forest types (columns in Figure 3). SVIs exhibited noticeable differences concerning the variation range, the short-term post-fire behavior (three years), and inter-annual fluctuations. A wider range of values was observed in NDMI, NBR, and FRI2 compared to NBR2 and NDVI. The short-term post-fire behavior of normalized SVIs (NDVI, NDMI, NBR, and NBR2) displayed a sharp recovery, whereas FRI2 highlighted a more constant recovery rate through time. Inter-annual fluctuations of NDVI, NBR2, and FRI2 were less pronounced compared to those of NDMI and NBR. Recovery patterns of study areas RF and RM were similar through the entire time series as observed between study areas NA and LA (Figure 3a). Patterns of temporal trajectories at different burn severity classes highlighted that forest spectral recovery in the low and moderate class was generally at an advanced stage at the end of the time series (Figure 3b). Temporal trajectories at different forest types highlighted that broadleaved-dominated stands achieved slightly higher spectral recovery than those dominated by conifers (Figure 3c).

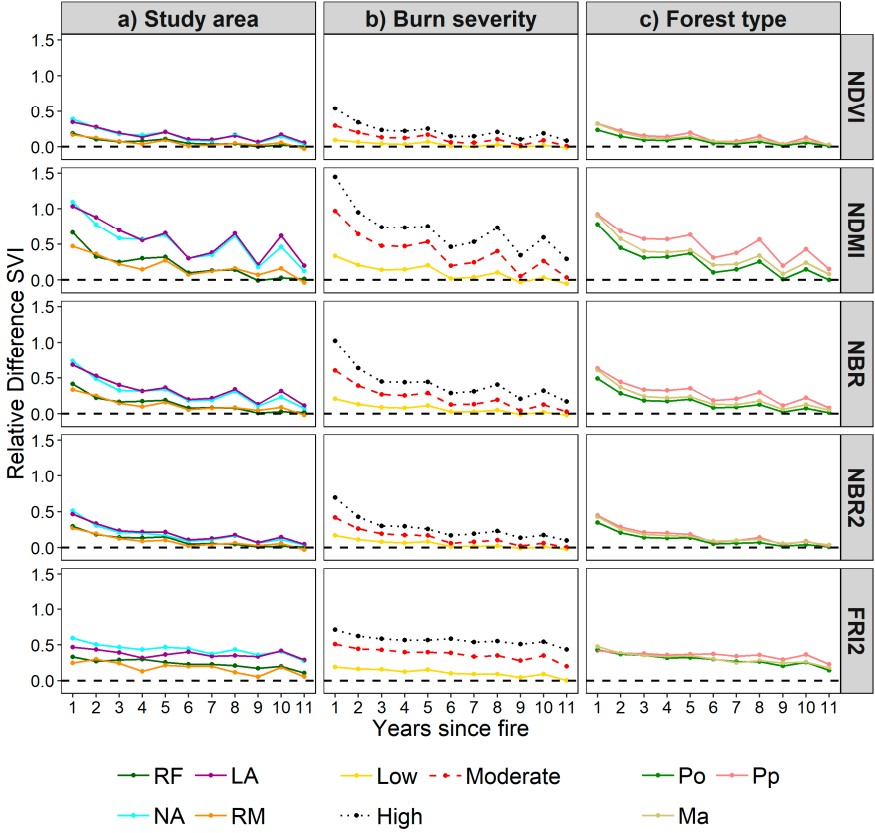

**Figure 3.** Post-fire temporal trajectories of Relative Difference SVIs (RDSVIs) (*y*-axis) at one-year intervals (*x*-axis) divided by study area (**a**), burn severity (**b**), and forest type (**c**). Years are expressed relative to fire occurrence. Study areas: Roccafluvione (RF), L'Aquila (LA), Navelli (NA), and Roccamorice (RM). Forest types: Po (pubescent oak), Pp (pine plantations), and Ma (manna ash and European hop-hornbeam).

### 3.3. Percentage of SVI Recovered Pixels

The percentage of recovered pixels for each SVI was computed considering pixels that completely recovered their pre-fire values by the 11th year after fire occurrence. They were aggregated by study area, burn severity class (Figure 4a), and forest type (Figure 4b). On average, NDVI displayed the

highest percentage (38.26%) compared to NDMI (28.83%), NBR (27.54%), NBR2 (26.23%), and FRI2 (24.92%). Differences between NDVI and the other SVIs increased for burn severity classes. In the high-burn-severity class, NDVI recovered 12.34% of pixels, whereas the average over NDMI, NBR, NBR2, and FRI2 was 5.5% (Figure 4a). Moreover, FRI2 displayed a larger separation between the recovered pixel percentage in the low class compared to the moderate and the high classes. Notably, the percentage of NDVI recovered pixels in Pp forests was higher (35.35%) compared to the mean value of the other SVIs (19.96%). The comparison between burn severity classes highlighted relevant differences of recovered pixel percentages, since the SVI averages were 50.36%, 29.42%, and 6.87% in the low, moderate, and high class, respectively. Within forest types, averaged recovery percentage of Po (32.03%) was higher than the averages of Pp (23.04%) and Ma (18.67%) (Figure 4b).

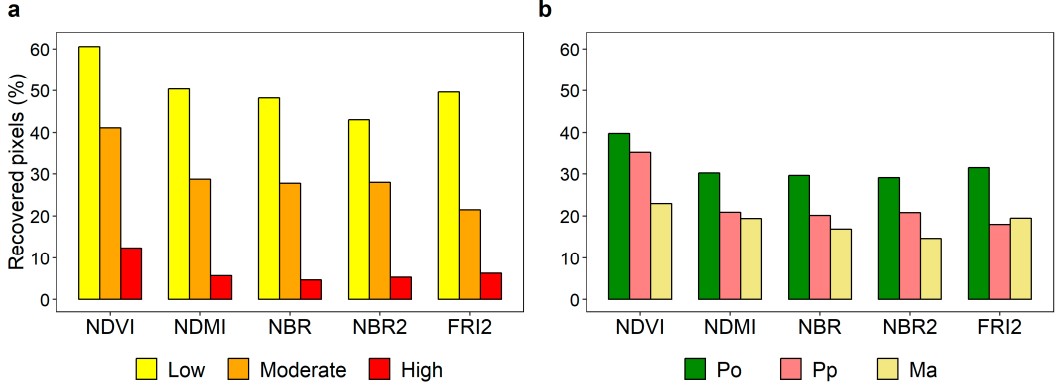

**Figure 4.** Percentage of recovered pixels aggregated by burn severity class (**a**) and forest type (**b**). Po: pubescent oak; Pp: pine plantations; Ma: mixed manna ash and European hop-hornbeam.

### 3.4. Long-Term Trends of RDSVIs

Long-term forest spectral recovery was evaluated through the median value of time required to attain pre-fire spectral conditions (Table 3) and by assessing the proportion of negative ($\tau < 0$) or positive ($\tau > 0$) significant trends ($p < 0.01$) to the total amount of detected trends at the pixel level (Table 4). The number of pixels displaying a significant negative simultaneous trend of all the SVIs was 2385 (214.65 ha). Among normalized SVIs, differences of the predicted recovery time length within each burn severity class and each forest type were very limited considering that the largest one was in pine plantations, and spanned over 1.27 years. The recovery time instead varied among different fire severity classes and ranged from 8.56 years of NDMI in the low-burn-severity class to 12.20 years of NBR2 in the high-severity class. As for forest types, years ranged between 9.74 for NDVI and 12.36 for NBR2 in Po and Ma, respectively. FRI2 instead required a longer time to attain spectral recovery compared to normalized SVIs. The difference in recovery time between FRI2 and the average of the normalized SVIs, expressed as the absolute value of years and as the relative difference, ranged from 1.3 years (14.7%) in the low-burn-severity class to 6.2 years (51.8%) in the high-burn-severity class. Among forest types, these differences varied from two years (19.5%) in Po to 4.9 years (45.9%) in Pp. The spatial distribution of Kendall's tau provided for each study area (Figure 5) depicts where recovering (green) or declining trends (red) were strictly monotonic and, thus, statistically significant for the MK test, depending upon each RDSVI. Considering those pixels exhibiting a recovery trend (negative tau), the percentage of significant ones was noticeably different both for SVIs and study areas (Table 4). Over the four study areas, FRI2 attained the lowest mean percentage of significant recovery trends (9.04%) compared to normalized SVIs (22.31%). In particular, study areas RM and LA exhibited the lowest percentage of significant recovering trends with FRI2, while this difference was less evident considering normalized SVIs. Moreover, FRI2 displayed a noticeable amount of pixels with a significant declining trend in the study area LA (5.6%) which was higher than those with a recovery trend (3.67%) (Figure 5).

**Table 3.** Median values and interquartile ranges (IQRs) of years required to complete forest spectral recovery within burn severity classes and forest types by pixels with a significant recovery trend of each Relative Difference spectral vegetation index (RDSVI) (*p* < 0.01). Po: pubescent oak; Pp: pine plantations; Ma: mixed manna ash and European hop-hornbeam.

|  |  | NDVI (IQR) | NDMI (IQR) | NBR (IQR) | NBR2 (IQR) | FRI2 (IQR) |
|---|---|---|---|---|---|---|
| Burn severity | Low | 8.99 (3.82) | 8.56 (3.68) | 8.85 (3.67) | 8.93 (4.19) | 10.13 (6.85) |
|  | Moderate | 9.88 (3.27) | 9.70 (3.23) | 10.03 (3.02) | 9.93 (3.75) | 13.38 (10) |
|  | High | 12.02 (2.53) | 11.68 (3.19) | 11.93 (2.39) | 12.20 (2.84) | 18.15 (14.13) |
| Forest type | Po | 9.74 (3.49) | 11.01 (4.42) | 10.59 (3.49) | 9.98 (3.99) | 12.35 (11.54) |
|  | Pp | 11.1 (3.25) | 10.08 (2.89) | 10.73 (2.80) | 11.12 (3.33) | 15.69 (11.39) |
|  | Ma | 12.18 (3) | 11.53 (3.56) | 11.85 (2.73) | 12.36 (3.44) | 15.58 (13.32) |

**Table 4.** Percentage of pixels with a significant negative trend (*p* < 0.01) of RDSVIs with respect to the number of pixels within the area of interest of the study areas. Study areas are Roccafluvione (RF), L'Aquila (LA), Navelli (NA), and Roccamorice (RM).

|  | Study Area | NDVI | NDMI | NBR | NBR2 | FRI2 |
|---|---|---|---|---|---|---|
| Percentage | RF | 26.86 | 36.48 | 39.88 | 38.42 | 18.24 |
| with a | LA | 19.49 | 16.04 | 21.97 | 27.92 | 3.67 |
| negative | NA | 14.87 | 18.9 | 26.99 | 19.06 | 10.9 |
| trend ($\tau < 0$) | RM | 7.56 | 10.65 | 17.07 | 14.87 | 3.35 |

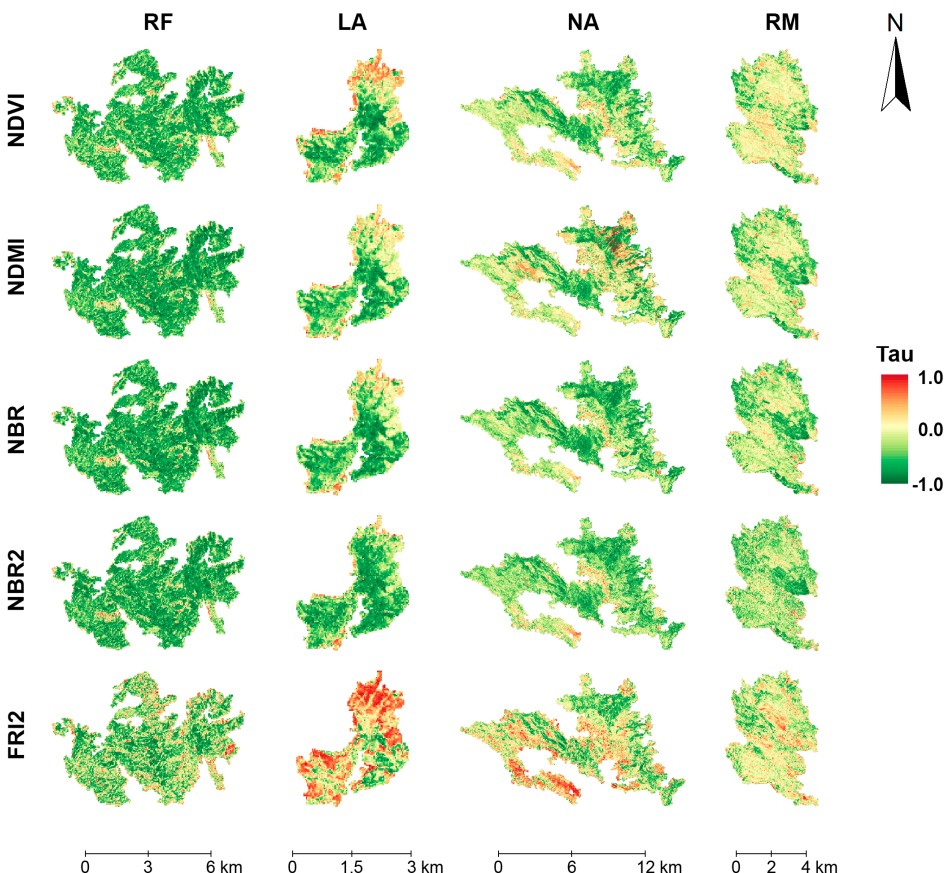

**Figure 5.** Maps of Kendall's tau rank correlation coefficient relative to RDSVI trends obtained with the Mann–Kendall test. Kendall's tau values range from 1 to −1. Positive values of tau are related to declining trends, whereas negative trends highlight spectral recovery. Study areas are Roccafluvione (RF), Navelli (NA), Roccamorice (RM), and L'Aquila (LA).

## 4. Discussion

*4.1. Main Differences between SVIs When Tracking Forest Spectral Recovery Dynamics*

Post-fire forest spectral recovery observed in four landscapes of the central Apennines is an ongoing heterogeneous process described by the decreasing patterns of the recovery metric (RDSVI) computed with all the SVIs. Nevertheless, substantial differences between the temporal patterns of the SVIs were observed. The different variation range between normalized SVIs is related to the sensitivity of each Landsat band to early post-fire changes [69,95] in terms of magnitude and direction. In this sense, post-fire variations were found to be higher in those SVIs contrasting one of the two SWIR bands and the NIR band (NDMI, NBR) compared to those using the Red and NIR bands (NDVI) or both the SWIR bands (NBR2) (Figure 3). The ability of normalized SVIs to track the rapid spectral recovery occurring soon after the fire seems related to their sensitivity toward non-arboreal vegetation dynamics rather than to tree canopy cover development, as observed in comparisons with IFZ [42] and FRI [41]. The inter-annual variability of NDVI, NDMI, and NBR was higher respect to that of NBR2 and FRI2. Since sun angle effects should be minimized by the compositing algorithm, which prioritized images acquired closer to a reference Julian day, inter-annual fluctuations seem more influenced by phenological and precipitation effects. As observed by Song et al. [17], phenology variations in young stands affect the red, the near-infrared, and the shortwave-infrared bands, which are the ones employed by NDVI, NDMI, and NBR. On the contrary, NBR2 seems less sensitive to variations due to precipitation effects [38] and FRI2 also minimized inter-annual fluctuations through yearly spectral statistics of dense forest cover, displaying a linear behavior through time. The higher percentage of recovered pixels achieved by NDVI at the 11th post-fire year compared to the other SVIs (Figure 4) confirmed its broad sensitivity to the amount of photosynthetically active vegetation (herbs, forbs, shrubs, and tree regeneration) [23,96]. On the contrary, SVIs based on the SWIR bands were characterized by a low percentage of recovered pixels, indicating their higher sensitivity both to fire damages on forest cover and to the following forest structure recovery as observed in several studies on post-fire forest recovery monitoring using optical remote sensing [23,24,34]. Results from field-based measurements indicated FRI2 as the most correlated with tree canopy fractional cover.

*4.2. Forest Spectral Recovery of Different Burn Severity Classes and Forest Types*

Although thresholds employed for burn severity classes were not meant to be used in these study areas, post-fire spectral recovery differences were clearly recognizable at increasing burn severity degrees. This distinction was observed both in the temporal trajectories of RDSVIs (Figure 3b) and in the pixel recovery percentages at the 11th post-fire year (Figure 4a). Also, these results highlighted that there was slightly more similarity between the recovery achieved at the moderate- and high-burn-severity classes by FRI2 compared to that attained with normalized SVIs. Among the latter, it was also observed that SWIR-based SVIs (NDMI, NBR, and NBR2) achieved considerably lower recovery compared to that of NDVI (Figure 4a). Since serious damages likely occurred to the over-story layers at the moderate- and high-burn-severity classes, this seemed to greatly influence the recovery of FRI2 and SWIR-based normalized SVIs. This result suggests that primarily FRI2 and secondly SWIR-based normalized SVIs are sensitive to the gradual development of tree canopy cover. Similar results were already observed through the correlation between field measurements of burn severity and several differenced (post-fire minus pre-fire) SVIs at multiple time intervals [42]. Differences observed in the recovery of SVIs between forest types 11 years after fire occurrence seemed to be mainly driven by the magnitude of spectral changes detected soon after the fires (Figure 3c).

*4.3. Forest Spectral Recovery Time Derived from Monotonic Trends*

Modeling trends of the SVI recovery metrics in a spatially explicit manner by coupling Mann–Kendall and Theil–Sen methods allowed for the investigation of the rate of change at those pixels with a monotonically decreasing trend over time. Spectral trends at those locations can be

confidently attributed to the development of a post-fire secondary succession [86,87,97]. In this study, normalized SVIs exhibited spectral recovery periods of 12 years or less. However, the periods required by FRI2 to recover were up to 50% longer than normalized SVIs, particularly in the high-burn-severity class. This suggested that FRI2 is more suitable for tracking long-term forest recovery which results in slow rates of spectral changes due to the re-establishment of pre-fire tree canopy cover (sensu Frolking 2009) [8]. The sensitivity of IFZ and FRI to the advanced stages of the forest succession was observed in other studies addressing post-fire forest recovery assessment through different SVIs [41,42] and using IFZ alone [43]. The results obtained from the prediction of spectral recovery time were partly in disagreement with those of recovery percentages (Figure 4) as, in this latter analysis, few differences between normalized SVIs were observed. This can be explained because the number of pixels used to predict spectral recovery time was 2385, equal to 3.4% of the number of pixels involved in the analysis (69,444). These pixels were selected applying two major constraints to the trends of the recovery metric of SVIs. It was required that the RDSVI trends were significant in the MK test ($p < 0.01$) and that this was concurrently true at the same location. Despite the limited number of pixels, this approach allowed for the comparison between the recovery time of SVIs integrating spatial information. The resulting number of pixels was likely influenced by the lower percentage of significant trends of FRI2 compared to that of normalized SVIs (Table 4). Several factors could have limited the percentage of FRI2 pixels having a significant recovery trend in the MK test. Among these factors, FRI showed sensitivity to the delay of post-fire mortality of damaged tree crowns [41]. Thus, a subtle decline in tree canopies throughout the analyzed period could produce significant declining trends of the FRI2 recovery metrics. This was particularly relevant in the LA study area (Figure 5), where the percentage of significant declining trends was 5.6% of all the forest burned pixels, which was slightly higher than the percentage of recovery trends (3.67%). Also, it is arguable that delayed mortality of tree crowns occurring during the analysis period produced a shift in the direction of the spectral changes, resulting in statistically non-significant trends of the FRI2 recovery metrics. These factors highlighted that temporal trajectories of FRI2 at early stages of post-fire forest succession are generally non-monotonic compared to that of normalized SVIs. Hence, it is advisable that the assessment of significant trends with the MK test at the pixel level be performed considering the advanced stages of forest recovery. Moreover, benefits could come from the use of a contextual approach, exploiting the information of neighboring trends to assess their monotonicity with the MK test [84].

## 5. Conclusions

Assessing post-fire forest regeneration dynamics by means of multi-temporal change detection analysis with Landsat imagery and SVIs allowed exploring different temporal scales of this process. In order to better estimate the future trajectories of forest recovery, it is crucial to understand which SVI can serve better to achieve this scope. Modeling changes of SVIs over a sufficient period with a robust regression approach can effectively address this matter. This study highlighted that the enhanced FRI2 ability to track long-term forest regeneration dynamics could be associated with ecologically meaningful results regarding the length of the forest recovery process and referring to the re-establishment of a continuous canopy cover over the burned areas. Therefore, the choice of the most suitable SVI for post-fire vegetation recovery assessment should be based upon the existing type of vegetation cover and the appropriate timescale. Early to medium stages of the post-fire forest secondary succession can be monitored using a normalized index employing the SWIR bands. However, at a time scale wider than 10–12 years, FRI2 provided reliable results through linear modeling extrapolation. Further research is needed to test its suitability at advanced stages of post-fire forest recovery.

**Supplementary Materials:** The following are available online at http://www.mdpi.com/2072-4292/11/3/308/s1: Table S1: Detailed list of the Landsat data used in this study; Table S2: Landsat images used to map burn severity in the four study areas: Roccafluvione (RF), L'Aquila (LA), Navelli (NA), and Roccamorice (RM).

**Author Contributions:** Conceptualization, D.M. and M.G.; data curation, D.M.; formal analysis, D.M.; funding acquisition, C.U. and M.G.; investigation, D.M. and A.V.; methodology, D.M. and M.G.; resources, C.U. and M.G.;

software, D.M.; supervision, C.U. and M.G.; writing—original draft, D.M., A.V., and M.G.; writing—review and editing, D.M., A.V., C.U., and M.G.

**Funding:** This research was funded by the University of Torino, research grant "Monitoraggio della rinnovazione forestale post-disturbo attraverso il telerilevamento satellitare e rilievi di campo" and by the Marche Polytechnic University, through the project "Aerial Drone for Environmental and Energy Research".

**Acknowledgments:** The authors wish to thank Francesco Malandra for his contribution to field data collection.

**Conflicts of Interest:** The authors declare no conflict of interest.

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
