# Peer review of "Forest Spectral Recovery and Regeneration Dynamics in Stand-Replacing Wildfires of Central Apennines Derived from Landsat Time Series"

_remotesensing, doi:10.3390/rs11030308_

Round 1
Reviewer 1 Report
The study provides an overall assessment of different spectral indices in monitoring post-fire forest recovery that can have significant scientific contribution as finding suitable remote sensing indices for monitoring post forest disturbance are still challenges. The manuscript was well prepared and appropriately presented with comprehensive data and results. The reviewer recommends the manuscript for publication on Remote Sensing journal.
Some comments should be considered and addressed in further steps as below:
Line 17, what is SVI - need to be detailed here
Line 24-26, what is implication for NDVI? is it still suitable for monitoring post-fire recovery? at which stage? specific to vegetation types?
Line 89: Typo "e"??
Lines 110-113, Latin should be in Italic? Check with Instruction for Author by Journal.
Figure 1 caption should include detail expression of RF, LA, NA & RM
Table 1 & Section 2.4, why Landscape metrics were included? I didn't see any application or explanation of these metrics in the study. Remove them if they were not used.
Line 192-194, you are using burn severity threshold from a study in North America that has much different in land cover type compared to your study area. How those burn severity thresholds can be reliable to use for your study area? Need to justify the application of these thresholds for your study or find other thresholds that was developed for your study area.
Line 220, please check the citation that not matched with the reference.
Figure 4 caption should include the meaning of tau values, i.e. > 0 indicates positive trend of spectral recovery, <0 indicates negative trend of spectral recovery.
Lines 414-415, please check citations that do not match with the contents of these two studies.
Author Response
We provided the point-by-point response to the reviewer's comments in the attached Word file.

Reviewer 2 Report
This is an interesting topic and the authors have clearly spent considerable time developing and testing the statistical methods against the satellite and in situ data. In general, this local- to regional-scale analysis is very promising but the narrative is at times hard to follow. This is a long paper and some editing down of figures and tables, maybe placing them in Supplemental Materials, is needed. Some additional English language editing work would be helpful. Specific comments are below.
Specific comments:
Lines 110 - 112, the descriptions of tree species and Po and Ma forest types (?) is confusing. Please rewrite. Including the scientific names much appreciated. Does that make the Pp another type of forest? Please make this clear at the beginning as the forest types are referred to later on in the paper. In general, this is a short Study Area description and more could be added to describe these past wildfire events as well as climate relative to other Mediterranean ecosystems.
Table 1 missing units for many of the rows but some rows have units. Please standardize for the rows that need units. Include abbreviation of landscape pattern metrics in the table caption.
Line 194 has an extra word 'sensu'. Digitization not digitation.
Why were the burn severity classifications from Miller et al. (2007) used (ref # 69)? Do the authors have some explanation why their study area in the Apennines are similar to the Sierra Nevada mountain range in California, USA?
Lines 270 - 282: How much field data was collected? This becomes more important in the Results section line 367. Is the 50,200 regenerating trees per hectare from the field data?
Figure 2 should be redone to make it easier to read. I suggest the authors move the legend from side to the bottom, whereby only the variables related to Area are under Area, etc. For example, Burn Severity column would have a legend underneath it with Low, Moderate, High (similar to Figure 3). As is, very hard to translate back and forth. Include spelled out names of abbreviation sin the figure caption.
Tables 3 and 4 should be reduce to 3 significant figures and not this combination of 3 and 4 significant figures. To be clear, the authors are saying these are the estimates at a p-value of < 0.01? So all are significant?
Figure 4 is very informative and should be enlarged. Explain in the caption more what the legend is saying about timing of recovery trends.
Lines 367 - 383: Provide more context for this result. Is this type of density and distribution of seeding and saplings low, expected, high, and/or somehow unique for the species reported? How were these calculated? A count across the burn types from field data or from the correlation? Please make this more clear. Figure 6 may be superfluous, given that it's findings are described in the text and there are few outliers.
In the discussion or conclusions, it would be a great addition for the authors to discuss how this work - specifically the veg and fire algorithms- might be extended given 20 m Sentinel-2 data.
Author Response

(The authors gave the same response as above.)

Reviewer 3 Report
This is an interesting study, but the paper is not ready for publication. The methods appear sound enough, however they lack focus and coherent structure. There are some parts that seem irrelevant and other parts that need more explanation/justification. In addition, the methods don’t flow logically from one step to the next, and the use of many different software packages and techniques seems unnecessarily complex for what is essentially a Landsat pixel-based analysis. There are many grammar and spelling errors throughout. The entire text needs to be edited by someone with advanced English skills. Some specific comments are as follows:
The abstract should clearly state the aims of the paper. The goal (line 12) is currently vague. Is it to compare indices, or track post-fire recovery? I would be more specific, such as “the goal was to determine which Landsat based spectral index is most accurate in tracking post-fire recovery…”
The acronyms in the abstract should be spelled out in full.
The main results in the abstract need to be clearly articulated, along with the significance of your findings.
The introduction is OK but could be more targeted towards what is specifically relevant to this study. For example, you mention the Vegetation Change Tracker (VCT), but the only reason is to introduce the Integrated Forest Z-score (IFZ), therefore VCT itself is irrelevant.
The hypotheses / objectives of the study need to be better articulated. The first one (a, on line 94) is a bit obvious. The second (b) is OK. The third (c), as it is currently written, is unclear to me. Also, the final sentence (line 99) does not fit logically with the other aims. It is not clear how the field data fits within the entire study. I think it would be better to use the field data as a validation tool for your remote sensing results.
Figure and table captions should allow the figure/table to be understood without having to refer to the main text (e.g. to find out what acronyms mean etc.)
In general, acronyms are over-used, in my opinion. Unless it’s a familiar acronym (e.g. NDVI), it makes it harder for readers and interrupts the reading experience. I would not abbreviate fire names or tree species (e.g. Lines 107-114). Even SVI is somewhat unnecessary, as you can just say indices, or spectral indices. Likewise with LTS.
Latin names (e.g. line 110) should be in italics.
Most of Table 1 doesn’t seem relevant, and I think can be removed. The climate variables are very similar for all fires and the fragmentation statistics do not seem to serve any purpose. In addition, the frag stats are not explained until much later in the text (Section 2.4).
Section 2.2 could be condensed. It is also a bit confusing in that you say you use LEDAPS for the USGS data and the 6S algorithm for the ESA data. The way it is written sounds like you’ve used two different surface reflectance methods. I am also unsure whether you created annual composites, or seasonal. You say seasonal, yet talk about DOY.
Section 2.3 is not clear. It seems that this is essentially about generating a forest/non-forest mask, however I would question the need for a 0.5m forest mask to use with 30m Landsat pixels. You also mention here that you merge these maps with burn severity maps, but the burn severity is not addressed until later (Section 2.5) so it is a little confusing. It also seems that you make a different forest mask later (line 217) using a Random Forests classifier, as opposed to the SVM used here. Why are there two different forest masks?
Section 2.4 seems irrelevant to this study
Section 2.5 needs more justification/explanation around the thresholds used (line 193). Also how is the dNBRoffset calculated?
Line 194 – what is sensu?
Section 2.6 – I would consider moving some of the information around spectral indices in the Introduction to this section, so that you clearly justify your choice of indices.
In line 207 Swir should be SWIR. I am also unsure what you mean here. Are you saying that you only used these three bands in your FRI calculation, or did you use all six optical bands?
Line 209 – what does “its algebric behaviour is comparable” mean? (also, it should be algebraic)
I am unsure what the yearly statistics of forest cover (line 217) are used for. Is this part of the FRI calculation?
Section 2.8 – I’m not fond of this title. Perhaps something like “Statistical analysis of recovery trends”
Line 246 – what does high breakdown point – around 29% mean?
Line 254 – Pre-withening should be pre-whitening (I believe) and needs to be more clearly explained and justified.
I am also wondering whether there is a need for such precise statistical procedures. Is there any practical benefits for using these techniques, when Landsat time series is such a crude estimation of forest recovery? Can you make a stronger case for using the Theil-Sen / Mann-Kendall / Pre-Whitening method over other simpler linear regression techniques?
I don’t understand lines 265-269. What do you mean by “inverse prediction” and “spatial intersection”?
Section 2.9 – How does the field data link to the rest of the study. I think you need to consider it as more of a validation tool. In the current paper it is almost like a separate study in itself.
I would like to see the results more closely aligned with the methods in terms of structure. The results also appear to contain information not explained in the methods, such as inter-annual phenological variations.
In Figure 2 the caption should explain the acronyms (fire names and tree species) and I think the scale of the y-axis needs to the changed so that differences in Forest Type are easier to distinguish.
The results in Table 3 concern me. I would expect NDMI, NBR and NBR2 to all be similar, but I would expect NDVI to produce a shorter recovery. I’m not familiar with FRI, but it seems extraordinary that it is producing time-frames that are so much longer than the other indices (2-3 times longer). If these results are valid then this is a MAJOR finding and should be clearly highlighted in the abstract/discussion/conclusion. (Perhaps check your calculations and consider whether your sample size is sufficient).
It is also concerning that the results in Figure 3, which clearly show NDVI recovering more quickly (as expected) and FRI behaving similar to the others, do not come across the same in Table 3.
Lines 335-336 – what are “Delta” values – these numbers (4.8 etc) are not in any table.
Table 4 – shouldn’t most pixels have a positive slope rather than negative? Given that they are recovering.
In Figure 4, one map (LA/FRI) seems quite different than the others. I think this should be discussed in the discussion. I am also wondering how you produced these maps when in your methods (section 2.7) you use pixel samples, rather than all pixels.
Lines 375-379 – why have you switched to full names here for tree species, instead of Po, Pp etc used previously? I prefer the full names, but either way you should be consistent.
As mentioned earlier, the field results (although interesting) don’t seem relevant to the rest of the study.
The discussion needs to more clearly link to the aims of the study, highlight your major findings, and compare your results to other studies. You need to focus more on communicating what your key messages are, whatever they may be (e.g. that FRI is preferable to the simple ratio indices, or that Theil-Sen is better than OLS regression). In addition, you should attempt to explain any unexpected results (e.g. that FRI is predicting recovery time-frames 2-3 times that of other indices).
Line 388 – Again, why are you discussing inter-annual variability, when it is not in your methods?
Line 470 (and in abstract) – I am not convinced by the statement that “only FRI provided reliable results”
The errors in grammar are too numerous to mention, and not the role of reviewers to fix. As an example, on line 429 you say “allowed to compare”, however you need to say “allowed us to compare” or “allowed for the comparison”. These kind of errors are throughout the entire manuscript.
Author Response

(The authors gave the same response as above.)

Reviewer 4 Report
This study focuses on prediction of forest recovery after wildfires using landsat time series.
It is an interesting topic and of great importance. But in my opinion, this paper is not ready to be accepted.
1. The paper was not well written, some sentences are very hard to follow. I had to guess what the authors meant to say.
2. According to line 106-107, the fire event occurred in July-Aug 2007, however, in Table 2, the in-situ forest/nonforest data of the sites RM and LA were acquired on 09/07/2007. Why did the authors collect the data during fire event?
3. Line 156-line 163: please clarify how the ROIs were chosen.What is the percentage of forest/non-forest for each site, also please provide the omission error and omission error for the classification in Table 2.
4. Line 184-186: I assume the authors used 2006 as the pre –fire year and 2008 as the post-fire year. But the fire happened in 2007, one year before 2008, a great part of the burn scar signals can be weakened by rain, wind as well as biomass recovery. Why did not the authors choose the images right before and after the fire event to estimate the burn perimeter and severity? These images before and after the fire may have different solar/viewing angles, but all the effects can be corrected. Also, the land cover in 2007 maybe different from the year 2006, how to prove that all the changes in eqn.(1)-(3) were caused by the fire in 2007?
5. The burn severity and perimeter were estimated with RdNBR, it is very hard to estimate burn severity and area with any single parameter. Line 194 stated that the fire perimeters were corrected through on-screen inspection. Have the authors performed any validation of the burn severity and perimeter estimation?
6. The surface reflectance data were obtained with different atmospheric correction algorithms. Have the authors compared different algorithms? If there were differences, what are the effects on the final results?
7. Line 237: eqn.8 how to decide the value of the offset? Why the prefire median was obtained using the data from 2001-2007? The fire was in 2007, the condition in 2007 cannot be regarded as prefire.
8. Line 249-269: please provide more details about the equations.
9. The images from different landsat sensors were used. However, the spectral response and band width of landsat sensors are not the same, how did the authors solve this issue? Or can the authors show the effects of the spectral response and band width are small enough to be neglected?
Minor comments:
1. please provide the exact dates of the fire and the acquisition dates for all the landsat images so that the readers can know which images can be regarded as pre-fire and post-fire.
2. What is the source of the data in Line 108-109, the area influenced by fire.
3. What is the size of each study area?
4. Line 157: “larger than 0,5 ha”, should it be 0.5 ha?
5. Line189: eqn (3), what is 0,5?
Author Response

(The authors gave the same response as above.)

Round 2
Reviewer 2 Report
This version is an improvement. Thank you to the authors for providing a revised and much improved manuscript.
Author Response
We provided a reply to your comments in the attached Word file.

Reviewer 3 Report
The paper is much improved and the authors have done well in addressing the reviewer comments in a short period of time. I consider it worthy of publication subject to the following:
On line 89, change ‘is mostly’ to ‘can be’ and remove ‘mostly’ from line 90.
On line 94, change ‘at comparing’ to ‘to compare’
Line 101. Hypothesis (c) is not a testable hypothesis as it is currently written.
Section 2.2. The use of the word ‘seasonal’ is confusing, as to me that means 4 images per year (1 in summer, 1 in autumn, etc.) I would prefer the word ‘annual’ or ‘summer’ was used.
Table 2 does not seem necessary (or could be moved to supplementary materials)
Line 177. I would prefer this was written more along the lines of “The thresholds defined by Miller and Thode [67] were used to classify burn severity…”
Line 186. FAO should be written in full
Line 222. What forest mask are you referring to here?
Line 228. Change ‘variable’ to ‘variables’. And delete the word ‘ones’
The explanation behind the various forest masks is still a little unclear in the manuscript. It is not clear what the CLC forest cover map (line 112) was used for. Then in Section 2.5 (around lines 224) you should explain more clearly why you used random forests to make another mask. (e.g. “because the mask created in Section 2.5 only covered pixels within the fire perimeter, we used these results to create a Random Forests model in order to classify areas outside of the fires”)
245. change ‘pixel-wise median’ to ‘The median pixel value’
Line 252. You have two equation (8)s. This should be equation (9) – therefore all the remaining equations in the text need to be altered.
Section 2.8. I think you need to make a stronger case for using Mann-Kendall and why you think there is serial correlation in your data. I agree that Theil-Sen is a useful procedure because it is more robust to outliers, however it is not obvious the benefits that Mann-Kendall is providing. You expect there to be spectral recovery after a fire, so whether the trend is ‘significant’ may not be relevant. If a pixel recovers in a hap-hazard way it may not be as significant (according to MK) as one that recovers in a steady fashion, but it is still recovering. You also only have 8 or so years of data, and MK is less reliable with few data points. These limitations could be addressed in your discussion.
Line 285. Remove the word ‘lag’ and change ‘recover’ to ‘return to’
Section 3.1. You should make it clear that you are comparing ‘field based’ data with ‘Landsat-derived’ SVIs
Line 350. What do the values in brackets represent? E.g. (14,66%)
Line 414. It is unclear how this reference [69] ‘seems to confirm’ FRI2 is more suitable
Line 428. Is FRI2 more ‘reliable’ or more ‘sensitive’?
Line 434. Are you saying that only 3.46% of pixels overall had a ‘significant’ trend? If so, this suggests Mann-Kendall may not be all that useful in your study.
The discussion section overall is still a little unclear, particularly section 4.3. It is sometimes not obvious whether you are discussing your results, or those of other studies. You also could consider more explicitly discussing the limitations of your methods and where future opportunities lie.
There remain quite a few sentences that contain grammar errors, or which are confusing due to how they are worded. Note that in many cases, a comma is all that is needed. Or alternatively, long sentences should be split up. For example, lines 40, 46, 47, 51, 81, 107, 120, 124, 128, 151, 154, 156, 250, 256, 261, 264, 299, 320, 330, 344, 352, 353, 379, 399, 408, 418, 428, 432, 446, 448, 451
Author Response

(The authors gave the same response as above.)

Reviewer 4 Report
The authors addressed all my concerns
Author Response

(The authors gave the same response as above.)
